# Adult Neurogenesis in the *Drosophila* Brain: The Evidence and the Void

**DOI:** 10.3390/ijms21186653

**Published:** 2020-09-11

**Authors:** Guiyi Li, Alicia Hidalgo

**Affiliations:** School of Biosciences, University of Birmingham, Edgbaston, Birmingham B15 2TT, UK; G.Li.2@bham.ac.uk

**Keywords:** *Drosophila*, neurogenesis, gliogenesis, brain, adult, cell proliferation, BrdU, EdU, FUCCI, PCNA, MARCM, stg, neuroblast, neural stem cell, progenitor, dMyc, miR-31a, MyD88, wek, Toll-2, Yki, eiger, TNF, inscutable, deadpan, plasticity, homeostasis, injury

## Abstract

Establishing the existence and extent of neurogenesis in the adult brain throughout the animals including humans, would transform our understanding of how the brain works, and how to tackle brain damage and disease. Obtaining convincing, indisputable experimental evidence has generally been challenging. Here, we revise the state of this question in the fruit-fly *Drosophila*. The developmental neuroblasts that make the central nervous system and brain are eliminated, either through apoptosis or cell cycle exit, before the adult fly ecloses. Despite this, there is growing evidence that cell proliferation can take place in the adult brain. This occurs preferentially at, but not restricted to, a critical period. Adult proliferating cells can give rise to both glial cells and neurons. Neuronal activity, injury and genetic manipulation in the adult can increase the incidence of both gliogenesis and neurogenesis, and cell number. Most likely, adult glio- and neuro-genesis promote structural brain plasticity and homeostasis. However, a definitive visualisation of mitosis in the adult brain is still lacking, and the elusive adult progenitor cells are yet to be identified. Resolving these voids is important for the fundamental understanding of any brain. Given its powerful genetics, *Drosophila* can expedite discovery into mammalian adult neurogenesis in the healthy and diseased brain.

## 1. Introduction

Whether neurogenesis occurs in the brains of adult humans and other animals, and to what extent, has long been debated and controversial. Were it to occur, it would transform our understanding of how the brain works. It would imply that neural circuits are not static, and instead can be modified and elaborated, as glia and neurons are added throughout life. It would provoke the question of whether adult neurogenesis is required for normal brain function, and whether it is impaired as we age. It would transform strategies for tackling brain damage and disease, as it would open the opportunity to restore neural circuits and function. Acquiring evidence of adult neurogenesis is technically challenging. Both in mammals (including humans) and in invertebrates, evidence has generally relied on cell cycle and lineage tracing markers, such as 5-Bromo-2-deoxyuridine (BrdU), and neural stem cell markers [1,2,3,4,5,6]. However, the ability to detect these markers could depend on exactly how experiments were carried out, and as a result, disparate findings have been feeding the controversy [1]. Importantly, adult neurogenesis in humans seems limited to discrete sites (e.g., hippocampus, involved in learning and memory), but this may not be the case throughout the animals (e.g., in fish it is more widespread). Whether spatial differences in the patterns of neurogenesis have functional implications is an intriguing question.

Adult neurogenesis in the *Drosophila* brain has been debated for over three decades. One original argument against it was the presumption that invertebrate brains would be ‘hardwired’, but this was ruled out by the abundant evidence of plasticity in the nervous system of *Drosophila* and other insects. Instead, a valid argument against adult neurogenesis in *Drosophila* has long been that developmental neural stem cells—called neuroblasts in *Drosophila*—are eliminated before adult flies eclose. In the absence of neural stem cells, it is unclear how adult neurogenesis could proceed. Despite this, evidence is accumulating that cell proliferation can take place in the *Drosophila* adult brain. Furthermore, such adult cell divisions give rise to both neurons and glia. And neuroblast markers have also been detected in the adult brain. Thus, evidence indicates that adult neurogenesis can occur in the *Drosophila* brain. To look deeper into this, here we review the findings on adult neurogenesis in the *Drosophila* brain. We present the evidence for and against adult neurogenesis. We compare and discuss technical differences between studies, which led to contrasting results. And we discuss current challenges and the search for definitive evidence of adult neurogenesis. As a powerful genetic model organism, establishing and cementing whether adult neurogenesis occurs in *Drosophila* is fundamental to discovering fundamental principles, cellular and molecular mechanisms of how any brain works. It is also important in order to define the power and limitations of using fruit-flies in this context.

## 2. Developmental Neuroblasts Disappear by the End of Pupal Life

In *Drosophila* development, neurogenesis in the central nervous system (CNS) occurs in three stages: embryonic, larval and pupal. Embryonic neural stem cells, called neuroblasts (NBs), contribute 10% of adult CNS neurons, whereas post-embryonic larval and pupal NBs generate 90% of adult neurons [3,7,8]. Glia can originate both from neuro-glioblasts, that produce both neurons and glia, and glioblasts, producing only glia. We use the term NB to include both those that produce only neurons and also neuro-glioblasts. There are many excellent reviews on NB development (e.g., [8,9,10,11]). Here, we focus on their demise.

During embryogenesis, NBs are specified in the neuro-ecotoderm by pro-neural induction, the combination of Notch signalling, SoxNeuro and Worniu, and a coordinate-code of transcription factors [9,12]. NBs divide asymmetrically, to produce a self-renewing NB and a ganglion mother cell (GMC), which divides once to produce either two neurons, or a neuron and a glial cell [3]. Most embryonic NBs enter a reversible quiescent state at the end of embryogenesis and are reactivated during larval life by multiple factors, including nutrition [7,8,13].

In the larval CNS, there are multiple types of NBs, according to their region of origin: NBs of the thoracic and abdominal ventral nerve cord (VNC), optic lobes (OL), central brain (CB) and mushroom bodies (MB) [8,10,14,15](Figure 1). All larval NBs are of embryonic origin, except for the OL-NBs. These originate from pro-neural induction of the larval optic lobe neuro-epithelium, to form the outer proliferation centre (OPC), which produces medulla neurons, and inner proliferation centre (IPC), which forms distal cells connecting to medulla and lamina, and neurons of the lobula and lobula plate [16]. Some IPC NBs are induced following a migratory phase [16]. By contrast, lamina neurons originate from induction by retinal axons of lamina precursor cells (LPCs) [17,18,19]. NBs can divide following distinct profiles [10,11]. Type 0 NBs divide to self-renew and directly produce a neuron daughter cell. Type I NBs divide asymmetrically to self-renew and generate a GMC, which divides symmetrically once to produce either two neurons, or one neuron and one glial cell. Type I NBs comprise Type-IA NBs in the abdominal neuromeres and Type-ID NBs in the thoracic neuromeres, CB and OPC [8,9,11,20]. Type-II NBs of the CB generate intermediate progenitors (INPs), which first divide symmetrically amplifying their pool and then asymmetrically to self-renew and produce GMCs, considerably expanding their cell lineages [10,14,15]. Type III NBs of the optic lobe IPC, divide first asymmetrically to generate distinct NB types, and then symmetrically into terminally differentiated neurons [21]. All Type I and II NBs express *deadpan (dpn)* and *worniu (wor)*. Type I NBs also express *prospero (pros), miranda (mira), asense (ase)* and *grainhead (grh)*, Type II NBs do not express *pros* or *ase*, INPs do not express these initially, but subsequently switch to becoming Ase^+^, and Type III NBs express *atonal (ato)* [8,11,21,22,23]. The cell lineage each NB produces is regulated in space and time, by cascades of transcription factors that control neuronal identity and the timing of cell proliferation [9,10]. The specific genes involved in temporal cascades varies with NB type, but they all share the fundamental principle that as a result, the potential of NBs to give rise to distinct progeny cell types decreases over time [10,11]. All developmental NBs are subject to temporal cascades, and eventually, they are eliminated either through a last division that drives cell cycle exit and cell differentiation, or through the induction of apoptosis [24,25,26,27,28,29]. Type-1A NBs terminate cell division during larval stages, and MB-NBs stop proliferating last, in mid-pupal stages [8,27]. In between, different NB lineages terminate proliferation at different time points [8,25]. Transcriptional temporal cascades drive the end of NB life by switching to a cell state characterised by the down-regulation of early factors, up-regulation and nuclear translocation of Pros, reduction in cell size, cessation of cell proliferation and terminal cell differentiation [10,24,25,26,29].Type-I NBs of the VNC (ID), central brain and optic lobes (OPC), Type II NBs and their INPs, and Type III NBs, are all eliminated at mid-pupal stages through the upregulation of Pros and cell cycle exit [24,25,30]. By contrast, abdominal Type-1A NBs are eliminated earlier on in larval life, via Hox-dependent apoptosis [25,28,29]. This is caused by a burst of the homeotic protein Abdominal A, which intercepts the temporal cascade and activates downstream pro-apoptotic genes *reaper (rpr)*, *hid* and *grim* to induce cell death [25,28,29]. Some CB-NBs and all MB-NBs are eliminated by programmed cell death too. In *rpr* mutants, more than 70 neuroblasts in the pupal central brain persist until at least 30 h after puparium formation (APF), compared to less than 30 neuroblasts in wild-type pupae [27]. Similarly, MB-NBs disappear before 96 h APF in wild-type pupae, while in *rpr* mutants, all MB-NBs persist in the adult brain [27]. Thus, at least some CB-NBs and all MB-NBs are also normally eliminated by apoptosis [27] (Figure 1).

The elimination of NBs in pupae was traced using the combination of the NB marker anti-Dpn, the cell cycle E2F reporter Proliferating Cell Nuclear Antigen driven GFP (PCNA-GFP), which labels cells in S-phase, and the mitotic marker anti-phospho-histone-H3 (pH3) [27]. The number of Dpn^+^ NBs remained unchanged until 5h APF, and subsequently, NB proliferation, number and size decreased. By 30 h APF, a few CB-NBs and all MB-NBs remained. At 48 h APF, there were only MB-NBs left, at 96 h APF MB-NBs were hardly detected, and no NBs were detected in the pupal CNS from this time-point onwards [27]. Adult flies eclose at around 105 h APF, and by this time no developmental neuroblasts remain in the VNC or brain [27].

To conclude, evidence indicates that the adult brain does not contain developmental NBs. However, a lingering uncertainty is whether all INPs and their progeny cells have been traced, or whether conceivably some could remain undetected [14,15].

## 3. Experimentally-Induced Persistent Neuroblasts Divide in the Adult Brain

Interference with the normal developmental termination of NB divisions can cause over-proliferation that is sustained in the adult brain, for weeks. Using Mosaic Analysis with a Repressible Cell Marker (MARCM), *seven-up (svp)* mutant clones resulted in persisting Mira^+^ NBs that carried on proliferating in the adult brain [25]. Down-regulation of *pros, brain tumour (brat)* or *nerfin* in NBs expressing *chronologically inappropriate morphogenesis (chinmo), IGF-II-mRNA-binding protein (imp)* and *lin-28*, caused their continued proliferation in the adult resulting in massive brain tumours [31]. Blocking apoptosis and autophagy also caused developmental NBs to persist into the adult. In *rpr* mutant flies, MB-NBs persisted in the adult for at least 3 days. However, they still disappeared later, meaning that a Rpr-independent pathway also contributes to eventually eliminating supernumerary NBs [27]. Forkhead box O (FoxO), a transcription factor that promotes autophagy, is translocated to MB-NB nuclei at 72h APF in wild-type pupae, and also in persisting MB-NBs in *rpr* mutant flies. *FoxO* mutant adult flies, like *rpr* mutants, also had persisting MB-NBs [27]. And blocking autophagy by inhibiting ATG1 function, also delayed MB-NB termination [27]. The combined loss of function for both *foxO* and *rpr* resulted in MB-NBs that persisted for at least two weeks in adult brains [27]. Together, these data demonstrated that the combination of apoptosis and autophagy drives the elimination of MB-NBs. Importantly, persisting MB-NBs in adult brains did not form tumours, and instead produced neuronal progeny cells that sent projections along the MB lobes, potentially forming circuit connections [27].

Larval CB-NBs (Type I) that normally form the central complex can also be induced to persist through pupa and into the adult brain in a non-tumourigenic fashion. Simply reducing the levels of Pros with RNAi knodockdown in the *engrailed*-expressing DALv NBs, that normally generate interneurons of the central complex, resulted in supernumerary progenitors and neurons, in the absence of tumours [32]. Intriguingly, *pros* knock-down did not prevent neuronal differentiation, meaning that Pros is not required to induce neuronal differentiation, but to prevent GMC proliferation and the reversion of GMCs to a neural stem cell state [32]. Importantly, the supernumerary neurons could integrate into the ellipsoid body circuit, were functional, did not interfere with, and could contribute to, normal behaviour [32].

These extraordinary manipulations showed that the adult brain does not constitute an inhibitory environment that might prevent proliferation of NBs or the establishment of connectivity by newly formed neurons. Instead, supernumerary neurons can integrate into neural circuits and can function. This means that neural circuits in the *Drosophila* brain can accommodate variations in neuron number to deliver appropriate connectivity and behaviour.

## 4. There Are Proliferating Cells in the Adult *Drosophila* Brain

Adult neurogenesis requires that cell proliferation takes place, and the search for cell proliferation in the adult brain started almost four decades ago. Cell cycling is universally used as evidence of cell proliferation, as cells that terminally differentiate do not cycle. Post-mitotic cells are in G0, whereas cycling cells go through G1, S, G2 or M phases (Figure 2a). Cells can also remain quiescent in G1 or G2 for extended periods of time, prior to dividing, or they may never complete cell division. Grounded on these premises, ^14^C, ^3^H, BrdU and 5-ethynyl-20-deoxyuridine (EdU) incorporation and the E2F reporter PCNA-GFP are used to detect cells in S-phase, Fluorescent Ubiquitination-based Cell Cycle Indicator (FUCCI) to detect cells in all phases except G0, stainings against specific Cyclins for each phase, Cdc25/String for the G2/M transition, and pH3 for mitosis, by labs worldwide, in all model organisms (Figure 2a).

Technau used [3H]-Thymidine to detect DNA replication during S-phase, reporting that there were cycling cells in MBs of young adult female brains [33]. However, when BrdU incorporation, which also reveals DNA synthesis, was used to monitor the larval and pupal CNS, no cell proliferation was found after mid-pupal stages [3]. However, adult brains were not analysed. BrdU incorporation was tested specifically in the adult brain by feeding adult flies with food containing BrdU for 10 to 12 h [2], or for 24 or 48 h [27], and brains were fixed and analysed at different ages between 0–106 h post-eclosion. Still, no BrdU labelled cells were observed, leading to the conclusion that there are no proliferating cells in the adult brain [2,27]. These discrepancies were finally settled, using also BrdU incorporation experiments. When eclosed adult flies were fed with BrdU for 24 h, but in separate groups for each day, from day 0 to day 10 post-eclosion, up to 90% brains aged between day one to six contained at least one to three BrdU labelled cells per antennal nerve [34]. This frequency declined after day 6, although could still be detected by day 10. Similar data were obtained by feeding flies with BrdU for 3 h and dissecting either 6 h or 4 days later [35]. These data were confirmed with the BrdU analogue, EdU [36,37]. Eclosed flies were fed food containing EdU for 30 h, and brains were fixed two to six days later. EdU^+^ cells were found in wild-type brains, both at two days and six days [37]. Together, these findings confirmed that there are cells in S phase in the adult brain (Figure 2b). However, S-phase markers can also reveal polyploid cells, present in the adult brain [38].

The presence of cycling cells in the adult brain was confirmed using other G1, S-phase and G2 markers, PCNA-GFP and FUCCI, at the adult critical period (up to five days post-eclosion). PCNA-GFP^+^ cells were found in normal brains [6]. With FUCCI, degron fusion-proteins to tagged cell cycling proteins E2F-GFP and cyclin-B-RFP are degraded as cells enter S phase or G1, respectively [39] (see Figure 2a). Thus, FUCCI labels cells that are in G1, S, G2/M, or M/G1 phases of the cell cycle and does not label post-mitotic cells that are in G0 [39]. Control brains bearing the transgenes to visualise these markers but otherwise normal, also revealed the presence of FUCCI^+^ cells in G1/S, G2 and G2/M, potentially undergoing mitosis [6]. Presence of cells cycling through G2/M was also visualised using GFP-tagged Cdc25/String (Stg), whch is expressed in G2 and triggers the G2/M transition [6,40,41]. Accordingly, proliferating cells exist in the *Drosophila* adult brain, and most prominently at the critical period between 1 and 6 days post-eclosion [6,34,37].

There is evidence that cells do not only cycle, but complete cell proliferation. Firstly, BrdU pulse-chase experiments were carried out feeding adult flies with BrdU-containing food for two h, followed by fixing the brains either after 6 h or 5 days, to see if the number of labelled cells increased [34]. Since these labelled cells could only have emerged in the adult, any increase in cell number would be evidence of cell division. Indeed, whereas after 6 h most antennal lobes had only one BrdU^+^ cell, after five days antennal lobes had more than three cells [34]. These findings were reproducible [35]. Secondly, and similarly, following a 30 h pulse post-eclosion, the number of EdU^+^ cells increased between day two and day six, meaning that labelled cells proliferated after incorporating EdU [37]. Together, these data demonstrate that cell proliferation occurs in the normal adult brain (Figure 2b).

Ideal proof that cells divide would be seeing cells undergoing cell division, either in a time-lapse movie or with mitotic markers, such as pH3. But detecting mitosis in the *Drosophila* brain, in vivo, is difficult. It is the shortest phase of the cell cycle, cell cycles can be long and generally extend over the life-course, and mitosis might not occur synchronously in cell populations. In fact, none of the studies above reported mitotic markers or films, in normal wild-type brains.

A manageable alternative is to visualise mitotic recombination clones. These are based on the principle that recombination between somatic cells can only occur during cell division. Using MARCM clones [42], recombination is induced between FRT sequences upon conditional over-expression of Flippase, in the adult. The flies are heterozygous for GAL4 and GAL80, a repressor of GAL4, and therefore do not express GAL4 or any responding genes. Mitotic recombination causes the segregation of GAL80 to only one of the two daughter cells. In the presence of a reporter gene, e.g., GFP, the GAL80^−^GAL4^+^ cells are marked [42]. The advantage of this method is that once mitosis has occurred, the resulting daughter cells can be visualised at any time point in the life of the fly (so long as the cells remain alive). MARCM clones generated in pupae failed to reveal clones in adult flies [27], but this experiment did not test cell proliferation in adults. MARCM clones induced in the adult by heat-shocking for one hour in three-hour old flies, and visualising neurons with GFP a day later, did not reveal any clones [6]. However, cell divisions could have been missed with this protocol. Using one hour heat-shock in one day old flies, followed by three hours recovery, repeated four times per day, and followed by three days recovery, revealed βgal labelled MARCM clones in the antennal nerve of adult flies [34]. This meant that cell proliferation occurs in the adult brain. However, although heat-shock considerably increased the frequency of clones, clones were also present in control brains that had not been heat-shocked [34]. Similar results were obtained using double marked GFP^+^ RFP^+^ clones, where repeated heat-shocks between days 2 and 6 after eclosion increased the frequency of clones, but clones were also present in control brains [35]. Importantly, some of these clones were also BrdU^+^, demonstrating they were generated through cell division [34,35].

MARCM was refined to overcome the challenges imposed by unknown cell cycle length, its lengthening over time, stochastic and asynchronous proliferation. To bypass these, sustained *flippase* expression was introduced in Perma-Twin-MARCM [36], an adaptation of Twin-Spot-MARCM [43]. With Twin-Spot MARCM, GAL80 is not involved, and instead, flies are heterozygous for a genotype carrying GFP and RNAi towards RFP in one chromosome, and RFP and RNAi against GFP in the sister chromosome [43]. Upon Flippase-induced recombination at FRT sites, one daughter cell will express GFP only, as RFP is eliminated by RNAi, and the other daughter cell will express RFP only, as GFP is eliminated by RNAi [43]. With Perma-Twin-MARCM, *flippase* is expressed constitutively in all cells, under the control of the actin promoter, after tub-GAL80ts is turned off by shifting the flies to 29 °C [36]. Clones were induced: in 10 day-old flies, and dissected a day later; or just after eclosion, and allowed to grow at 29 °C for one, two, or three weeks [36]. Clones were generated in antennal lobes and optic lobe medulla, and the number of clones in each optic lobe increased as flies grew older [36]. However, about 25% of control brains that had not been heat-shocked also contained clones [36]. With RNAi-based clones [36,43], if RNAi does not completely eliminate the expression of GFP or RFP, the presence of reporters is no longer evidence of mitosis taking place. Altogether, evidence from MARCM clones is mixed. Nevertheless, Flippase consistently increased clone number compared to controls in the adult brain, and BrdU was detected in some clones [34,35], together suggesting that cell proliferation can occur in the adult brain.

Recently, Yorkie (Yki) was used to visualise proliferating cells in the adult brain [6]. Yki is a transcription factor and critical target of Hippo [44,45,46,47]. Hippo signalling phosphorylates Yki, causing it to be retained in the cytoplasm, thus inhibiting cell proliferation [44]. When this inhibition is over-come, Yki translocates into the nucleus, where it forms a complex with Scalloped (Sd) [44,45,47]. Together, Yki and Sd activate the expression of *E2F* and *cyclin-E* (both of which promote G1/S transition) and *stg* (which promotes G2/M transition), driving cell proliferation [41,44,45,47,48]. Yki translocates dynamically between the cytoplasm and the nucleus to promote cell proliferation [47]. Using a Yki-GFP fusion protein [46], Yki was found both in the cytoplasm, and nuclei of control adult brains [6]. Nuclear Yki-GFP was found in multiple brain regions, including the optic lobes, sub-aesophageal ganglion and central brain areas during the adult critical period [6]. As mentioned above, Stg-GFP [40,41] was also found in the adult brain [6]. Since both Stg and Yki are well known to provoke entry into mitosis, these data show there are proliferating cells in the adult brain.

## 5. Injury, Neuronal Activity and Genetic Manipulations Induce Further Cell Proliferation

Despite the above evidence, proliferation in the adult brain is rather limited and constrained. The total number of proliferating cells in the adult brain is not known, but the above experiments indicate that dividing cells could be a rather small fraction. Some of the detected cells may cycle, but could arrest at G1 or G2, including quiescent progenitors. Importantly, neuronal activity, injury and genetic manipulation can dramatically increase the incidence of cell proliferation.

Stabbing injury in the adult central brain increased both apoptosis and cell proliferation, as detected with BrdU, compared to non-injured controls [34]. Naturally occurring programmed cell death triggered adult cell proliferation in the adult, and upon injury, suggesting that these were homeostatic cell divisions [34]. Furthermore, these cell divisions required the TNF-α homologue Eiger [34,48]. Stabbing injury was also carried out in adult optic lobes in combination with clonal analysis with Perma-Twin-MARCM [36]. Flies were kept at 18 °C until seven days after eclosion, then the optic lobes were stabbed and flies shifted to 29 °C for two or nine days. Clones were observed surrounding lesions in both groups of flies, but those kept for nine days had significantly more clones [36]. Injury also triggered the nuclear localisation of Dpn [36]. Thus, stabbing injury induces proliferation in the adult brain. The proliferative response to injury in the central brain no longer took place in flies older than 10 days [34], but Perma-Twin clones continued to increase in optic lobes two weeks later [36]. Interestingly, injury resulted in the upregulation of the dMyc proto-oncogene in Dpn^+^ cells around the lesion [36]. Conditional overexpression of *dMyc* induced Dpn^+^ pH3^+^ cells, demonstrating that dMyc can induce progenitor cell proliferation in the adult brain [36]. Importantly, injury increased the incidence of MARCM clones compared to controls. Altogether, injury drives regenerative cell proliferation in the adult brain.

Loss of function for the micro-RNA *miR-31a* caused glial proliferation in the adult [37]. In *miR-31a* mutants, Repo^+^ glia cells were lost through apoptosis by day seven compared with the control, however, by day 21, the number of glia cells had recovered. Although the recovery was not complete, this indicated that proliferating cells compensated the glial loss caused by *miR-31a* loss of function [37]. In fact, more EdU^+^ cells were observed than in controls, meaning that glial apoptosis induced compensatory glial proliferation. To identify the source of miR-31a, a microRNA sponge was over-expressed in glia, neurons and neuroblasts. Glia were lost only when miR-31a sponge was driven in neuroblasts, with either *inscutable-GAL4 (insc-GAL4)* or *worniu-GAL4 (wor-GAL4),* and this also prevented the compensatory recovery of glial cells [37]. Using MARCM clones with *miR-31aGAL4*, the resulting progeny cells were both neurons and glia [37]. Together, these data showed that *miR-31a* is most likely expressed in adult progenitor cells, and it is involved in a homeostatic mechanism that maintains appropriate cell number in the brain throughout adult life [37].

Conditional activation of Toll-2 signalling in the adult brain increased cell number in the optic lobe medulla and central brain [6]. Activating neurons also increased cell number in medulla, and this was rescued by knocking-down *Toll-2* expression [6]. This meant that the brain is plastic and cell number can be modified by brain function. The increase in cell number was reproduced by manipulating the adaptors of Toll signalling, including downregulating *MyD88* and over-expressing *weckle (wek)* [6]. Furthermore, knocking down either *wek* or *yki* rescued the cell number increase caused by *Toll-2* over-expression [6]. MARCM clones could be induced by over-expressing *Toll-2,* with a brief 1h heat-shock, implying that *Toll-2* gain of function induces cell proliferation in the adult brain. This was confirmed with multiple cell cycle markers, which showed that the number of cells in G1 and M/G1 as seen with PCNA-GFP and FUCCI, and in G2 and G2/M as seen with FUCCI, Stg-GFP and nuclear Yki-GFP, all increased with conditional *Toll-2* over-expression in two-day old adults [6]. Remarkably, conditional over-expression in the adult brain at the critical period, of either *Toll-2* or *wek,* increased brain size, and conditional *yki* RNAi knock-down, rescued the increase in brain size caused by *Toll-2* over-expression [6]. Altogether, these data showed that brain function and Toll-2 signalling can promote cell proliferation in the adult brain via Yki downstream, and that this modifies brain size.

To conclude, the fact that neuronal activity, injury and gene expression can induce cell proliferation in the adult brain shows that the adult brain is plastic (Figure 2b and Table 1). A molecular mechanism senses interference with the status quo, and it can induce plastic growth and/or homeostatic adjustments. Intriguingly, plastic and homeostatic changes could have the potential to restore structural integrity and function—e.g., for regeneration—and potentially modify behaviour.

## 6. Gliogenesis and Neurogenesis in the Adult Brain

The evidence of cell proliferation in the normal adult brain raises two crucial questions: what types of cells divide, and what kinds of daughter cells do they produce?

There is clear evidence of gliogenesis originating from glia (rather than neuro-glioblasts). Firstly, BrdU^+^ cells in the adult brain were often Repo^+^ [34,35]. Secondly, apoptosis that was either naturally occurring or induced by either injury or *miR-31a* depletion caused glial proliferation [34,37]. Thirdly, compensatory glial cell proliferation continued in three-week old adult flies (although these glia could originate from NBs) [37]. Altogether, glial cells continue to proliferate in the adult brain and can homeostatically regulate their number throughout life.

There is also evidence of neurogenesis in the adult brain. Firstly, Perma-Twin mitotic recombination clones generated Elav^+^ progeny cells, suggesting that neuronal daughter cells were produced, and the incidence of neuronal clones increased with injury and over-expression of *dMyc* [36]. Secondly, over-expression of *Toll-2* in MARCM clones resulted in neuronal progeny cells, that formed axonal and dendritic projections that differentiated and targeted appropriately [6]. Thirdly, MARCM clones from *miR-31a-GAL4* cells resulted in both glial and neuronal daughter cells [37]. Fourthly, when the cell-lineage tracer G-TRACE was over-expressed conditionally in the adult with *tubGAL80ts,* with the neuroblast driver *insc-GAL4,* this resulted in progeny cells whose number increased during the subsequent seven days [37]. Amongst the resulting clones of progeny cells, 20% had Repo^+^ glia and 50% had Elav^+^ neurons [37].

Altogether, these data show that there is both gliogenesis and neurogenesis in the adult brain (Table 1). This is more frequent in the first seven days of adult life, but it can continue throughout adult life.

## 7. Touching the Void: What Are the Adult Progenitor Cells?

Glia can divide symmetrically to produce glial-only cell progeny, and this can suffice to explain at least some of the homeostatic divisions of glia. But glia can also be produced from neuro-glioblasts, and neurons are produced by NBs/neural stem cells. This raises the question of what kind of cells are the adult neurogenic progenitors?

Elav^−^Repo^−^ cells—i.e., lacking canonical *Drosophila* pan-glial and pan-neuronal markers—were recurrently reported in the adult brain [6,34,35,36,37]. Depending on the experimental conditions, they could account for 2–20% of proliferating cells in the adult brain [34,35]. Furthermore, no PermaTwin clones in the medulla were found to be Repo^+^ and instead where either Elav^+^ or Elav^−^Repo^−^ [36]. Elav^−^Repo^−^ cells could be progenitors or neural stem cells. Remarkable evidence for the presence of neural stem cells in the adult brain was reported using the NB driver *insc-GAL4*, as mentioned above. Switching on the lineage tracer GTRACE in *insc-GAL4* cells after eclosion, in the adult, revealed both Repo^+^ and Elav^+^ progeny cells, and cell number increased by day seven [37]. Since Insc is required for asymmetric division of NBs and both types of progeny cells were generated, this means that there are neural stem cells that divide asymmetrically in the adult brain.

What is the molecular signature of the adult progenitors? When NB markers Dpn, Mira, or Pros were used in adult brains, no such cells were originally found [27], but Dpn^+^ and Mira^+^ cells have since been reported [6,36]. Dpn was generally found in the cytoplasm under normal conditions, and became nuclear when optic lobes were injured [36]. Overexpression of *dMyc* also induced the nuclear translocation of Dpn together with the mitotic marker pH3, providing definitive proof that progenitor cells can divide [36]. Adult Dpn^+^ cells also express the adaptor of the canonical Toll signalling pathway, *myD88* [6]. *MyD88* is expressed in many cells in the adult brain, including neurons, glia and Elav^−^Repo^−^ cells [6]. All Dpn^+^ cells are also MyD88^+^ [6]. At least some MyD88^+^ cells cycle through G1/S and G2/M in the normal adult brain, and cell cycling is increased by activating Toll-2 signalling [6]. *Toll-2* over-expression induced cycling of Dpn^+^ MyD88^+^ cells [6]. Tolls can drive multiple signalling pathways downstream. In the adult brain, the Toll-2 dependent increase in cell number requires knock-down of MyD88, meaning that MyD88 normally keeps progenitor cells quiescent [6]. On the other hand, over-expression of *wek* increases cell number. Thus, under normal conditions MyD88 prevents cell division of adult progenitors, keeping them quiescent, while Wek can swing cells to proliferate when confronting stimuli [6]. The proliferation of MyD88^+^ progenitor cells downstream of Toll-2 involves the nuclear translocation of Yki [6].

Do these progenitor cells express other neural stem cell markers? It has been known for some time that the neural stem cell marker Eyeless (Ey)/Pax6 is present in cells of the adult brain [49]. Recent RNAseq analysis of the Drosophila central brain, optic lobes and whole brain [50,51,52] has evealed that NB genes are expressed in the adult brain (Table 2). This includes *ey, castor (cas), pox-neuro (poxn), chinmo, twin of eyeless (toy), dichaete (D), grainy head (grh), svp, mira, ase, wor, dpn, imp, lin28, numb* and *insc.* It also includes NB regulators such as *brat* and *zelda,* and genes involved in cell proliferation, such as *stg, yki* and *sd* [50,51,52]. Many of these genes encode transcription factors that could have pleiotropic functions in the adult [51]. Nevertheless, their expression also means that the adult brain has the genetic machinery ready to engage in neurogenesis and gliogenesis, were it to occur.

## 8. Seeing is Believing

The termination of developmental NBs by the end of pupal development and the technical challenges detecting proliferating cells buried the question of adult neurogenesis in controversy for three decades spanning the 20th century. This is not surprising, as detecting proliferating cells is very challenging. First, the length of the cell cycle is not known, cell cycle duration lengthens over time and adult life lasts over 30 days, making it difficult to know when to look. Secondly, cell cycling is a dynamic process and whereas static analyses can identify cells that are not post-mitotic, the fraction of cells in each phase of the cell cycle over time can be more informative. Dynamism affects proteins too. During cell proliferation, Yki shuttles in and out of the nucleus dynamically, rather than accumulating within nuclei [46,47]. Thirdly, mitosis is the shortest phase of the cell cycle and extremely difficult to detect. Still, seeing dividing cells is critical, because the methods reviewed above can have technical drawbacks. Capturing the exact moment a cell is in mitosis in the adult brain remains the dream result and an unsolved challenge.

An important argument feeding the controversy was that developmental neuroblasts disappear by mid-pupal development, precluding the presence of progenitor cells in the adult. However, the fate of all INPs has not been traced [14]. Furthermore, the argument assumes that adult progenitors originate from developmental neuroblasts, which may not be the case. In other insects, progenitors can originate from hemocytes [53]. In crayfish, neuronal precursors that give rise to adult neurons originate from haemocytes [53]. In zebra fish and mammals, glia cells de-differentiate to become neural stem cells [54]. Thus, progenitor cells in the adult brain could be different from developmental NBs.

## 9. Conclusions

The last two decades have provided evidence that gliogenesis and neurogenesis can take place in the adult *Drosophila* brain, and these increase with injury, neuronal activity and alterations in gene function. Some molecular mechanisms have been identified, and further progress in this context is anticipated. However, no cells have been caught actively dividing yet in the normal brain. Another critical remaining void is to discover the elusive, mysterious adult progenitors, what progeny cells they might produce and what circuits could be involved. Adult neuro- and glio-genesis endows the brain with structural plasticity and homeostasis, adjusting cell number upon genetic alterations, in response to injury, and in response to neuronal activity [6,34,36,37]. It has been argued that adult neurogenesis may endow the human brain with unique structural plasticity, enabling higher computation and the encoding of episodic memory [1]. Structural brain plasticity is linked to adult neurogenesis in *Drosophila* too [6]. Although the functional implications remain to be explored further, this suggests that structural plasticity and adult neurogenesis may be fundamental principles linking brain structure and function across the animals. The underlying genetic mechanisms could have also contributed to brain evolution. To conclude, the establishment of adult neurogenesis in *Drosophila* will expedite discovery into general principles of how the brain works, brain plasticity, homeostasis, and regeneration, with implications for understanding the human brain.

## Figures and Tables

**Figure 1 ijms-21-06653-f001:**
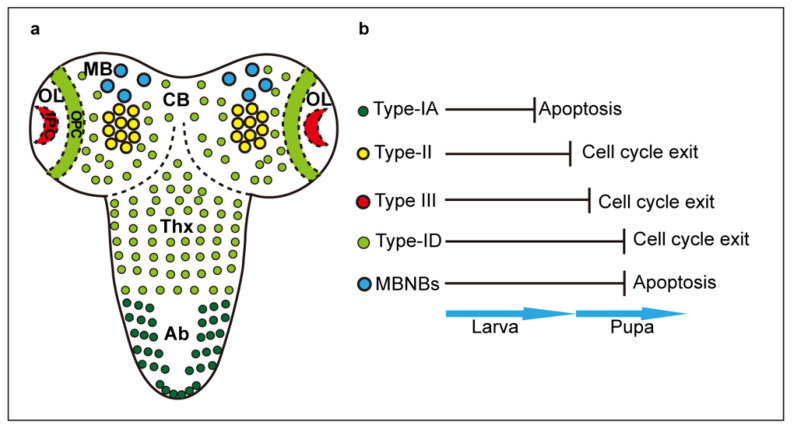
Neuroblasts in the larval CNS. (**a**) Distinct types and d istribution of developmental NB. (**b**) NB termination via cell cycle exit or apoptosis. CB: central brain; MB: mushroom bodies; OL: optic lobes; OPC: outer proliferation centre; IPC: inner proliferation centre; Thx: VNC thorax; Ab: VNC abdomen.

**Figure 2 ijms-21-06653-f002:**
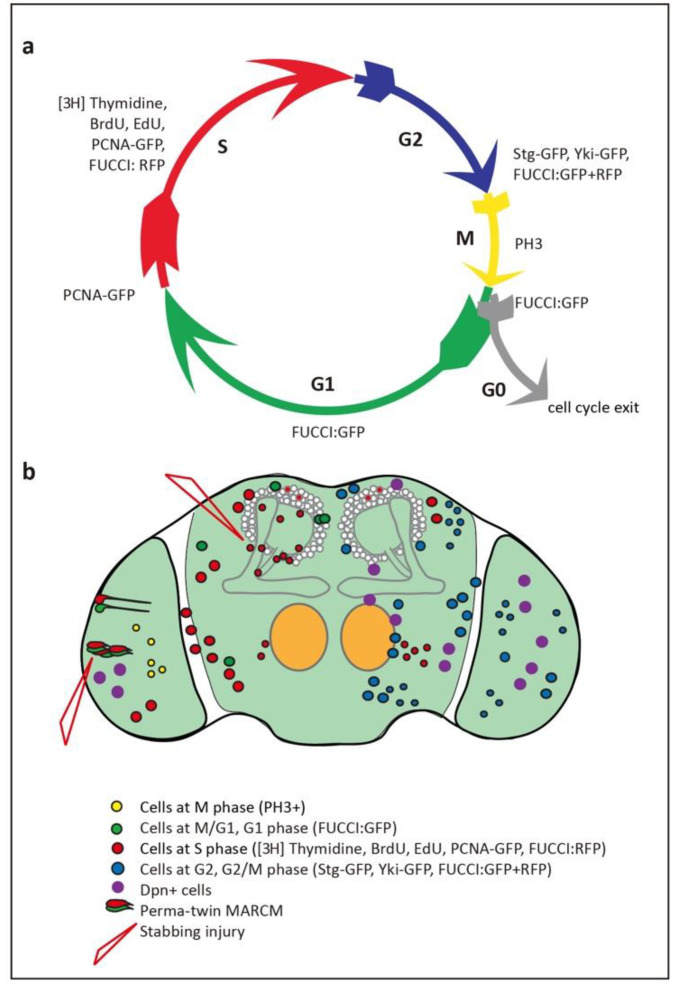
**Evidence of glio- and neuro-genesis in the adult *Drosophila* brain.** (**a**) Cell cycle phases and markers used to visualize them. (**b**) Summary of findings, indicating cells reported in each of the cell cycle phases in the adult brain, including in optic lobes, antennal lobes, sub-aesophageal ganglion and central brain.

**Table 1 ijms-21-06653-t001:** Evidence for and against adult neurogenesis in *Drosophila*. Summary of findings reported in the review, that either support adult neurogenesis or not.

Evidence of Adult Neurogenesis		Evidence Against Adult Neurogenesis	
Finding	Reference	Finding	Reference
Cell Proliferation			
**Cycling cells** detected in S-phase with ^3^H-Thymidine, BrdU, EdU, PCNA-GFP and FUCCI; in G1, with FUCCI; in G2, G2/M were revealed with FUCCI, nuclear Stg-GFP and Yki-GFP.	[6,33,34,35,37]	BrdU incorporation not detected in adult and PCNA-GFP was not seen after 96h APF.	[2,3,27]
		Polyploidy in the adult brain.	[38]
**Inference of mitosis from MARCM clones.** Clones induced in the adult brain generated both glial and neuronal progeny cells. Incidence of clones increased with flippase-induced recombination compared to controls. Some clones were BrdU+.	[34,35,36,37]	No MARCM clones detected in normal adult brains	[6,27]
		MARCM clones were detected in control brains that had not been heat-shocked, and Twin-Spot based approaches may not guarantee reporter knock-down	[34,35,36]
**Inference of mitosis:** A BrdU pulse in the adult resulted in multiple labelled progeny cells over time.	[34,35,37]		
Injury, Neuronal Activity and Altered Gene Function Can Increase Cell Proliferation			
**Injury** increased proliferation in central brain and optic lobes (BrdU, MARCM)	[34,36]		
**Altering gene function** can increase cell number, proliferation (various methods, including pH3) or brain size: dMyc, miR-31a, Toll-2, wek, MyD88, yki	[6,36,37]		
**Activating neurons** increases cell number	[6]		
Gliogenesis and neurogenesis			
**Gliogenesis**: Repo+ BrdU+ cells in MARCM clones, after injury, alterations in gene expression and lineage tracing of *inscGAL4* in the adult brain.	[34,35,37]		
**Neurogenesis**: Perma-Twin MARCM Elav+ clones, MARCM together with *Toll-2* over-expression and lineage tracing with *inscGAL4* in adult brain.	[6,36,37]		
Neuroblasts/neural stem cells			
Potentially unknown **Type II NB INPs** and progeny cells	[9,14,15,23]	Developmental neuroblasts are eliminated before adult eclosion	[24,25,27,28,29]
**Cells with NB markers** Dpn, Mira, Ey, *worGAL4* and *inscGAL4* in the adult brain. *InscGAL4* **with lineage tracing** in adult produced both neurons and glia	[6,36,37,49]	Dpn+, Mira+ and Pros+ cells disappear after pupa	[27]
**RNAseq** analysis revealed NB genes expressed in the adult brain	[50,51,52]	Typical NB genes can have pleiotropic functions	[50,51]
**Missing evidence**	Seeing dividing cells with pH3, other mitotic markers or time-lapse films Identification of adult progenitor cells, origin, model of cell division and resulting progeny cells		

**Table 2 ijms-21-06653-t002:** **RNAseq** analysis of the adult brain reveals expression of NB and cell proliferation genes. Summary of expression of NB and cell proliferation genes, from the databases by [50,51,52].

Gene	Number of Cells		
	CW-Midbrain ^1^	DA-Brain ^2^	KD-Optic Lobes ^3^
**cas**	6 cells	8 cells	24 cells
**d**	Many	Many	Many
**svp**	Some	Many	Many
**poxn**	A few	Some	24 cells
**hb**	Some	Many	A few
**kr**	Some	Many	A few
**grh**	27 cells	A few	Many
**toy**	Many	> Many	Many
**dac**	Many	Many	Some
**eyeless**	Some	Many	Some
**exd**	Some	> Many	Many
**br-c**	Many	Many	Many
**chinmo**	Many	> Many	> Many
**imp**	Many	> Many	Many
**lin28**	A few	Some	A few
**dpn**	1 cell	A few	A few
**wor**	4 cells	1 cell	13 cells
**mira**	11 cells	Some	A few
**ase**	8 cells	1 cell	8 cells
**numb**	Many	> Many	Many
**insc**	0 cells	19 cells	26 cells
**pros**	Many	> Many	> Many
**brat**	Many	> Many	Many
**zld**	Many	> Many	?
**yki**	Some	Many	Some
**stg**	12 cells	A few	A few
**sd**	Many	> Many	> Many
**KEY**	**A few**	30–200 cells	
	**Some**	201–1000 cells	
	**Many**	1001–10,000 cells	
	**> Many**	>10,000 cells	

^1^ CW [50]: Mid-brain, age not specified, 10286 cells; ^2^ DA [51]: Whole brain, 0–50 days old 56,902 cells; ^3^ KD [52]: Optic lobes, 3 days old, 57,601 cells.

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
