# Peer review of "Adult Neurogenesis in the Drosophila Brain: The Evidence and the Void"

_ijms, 2020, doi:10.3390/ijms21186653_

Round 1
Reviewer 1 Report
The work by Li and Hidalgo is comprehensive review of studies, which investigated adult neurogenesis in the adult Drosophila brain. The authors critically examine the evidence for and against adult neurogenesis. They also highlight technical difficulties in visualising and quantifying mitotic cells in the adult brain. It is an interesting review for researchers working in the field of neural development and adult neurogenesis.
Major points to address:
1) The introduction is rather short and should be extended to outline the content of the review. What was the purpose to write this review and who is the target audience? The authors could summarise in the introduction the key hypotheses and overall findings.
2) In line 37 references should be given for debate about human adult neurogenesis. For example Kempermann et al., 2018 Cell Stem Cells.
3) In parts the manuscript could benefit from a revised structure. Especially in section 2 it is not clear why certain neural stem/progenitor types are introduced and others are not. It makes sense to describe the different modes of divisions and terminations of Type IA /ID ,Type II and Mushroom body neuroblasts. However, it is not clear why Type III neuroblasts are mentioned and other stem cells types like OPC and IPC neuroepithelial cells, lamina precursors and others are not described. I would focus and describe in detail the types of precursors, the reader needs to know to understand the subsequent sections. Here, an additional illustration that shows the two main modes of termination (Apoptosis for Type IA, MB neuroblasts and terminal division for Type ID neuroblasts) would be informative. See for example Maurange and Gould, 2005, Trends in Neurosciences.
4) A crucial open question concerning adult neurogenesis in general could be discussed in a concluding section. Is adult neuro/gliogenesis in Drosophila functionally relevant for brain plasticity and repair? What do we know from other (invertebrate) model organisms ?
5) The manuscript could benefit from editing by a proficient English speaker. Structure and logic in language are not always coherent.
Minor points to address:
1) Line 28-35: should be removed.
2) Gene names should be written in full at the first time and subsequently abbreviated. For example see line 362.
3) In line 182 it should read:…..after incorporating EdU.
4) Line 191: Detecting mitosis in the adult Drosophila brain is difficult.
5) Line 392: Delete: 5.
Author Response
Reviewer 1
The work by Li and Hidalgo is comprehensive review of studies, which investigated adult neurogenesis in the adult Drosophila brain. The authors critically examine the evidence for and against adult neurogenesis. They also highlight technical difficulties in visualising and quantifying mitotic cells in the adult brain. It is an interesting review for researchers working in the field of neural development and adult neurogenesis.
We are grateful to this reviewer for the constructive suggestions that have helped improve our article.
Major points to address:
1) The introduction is rather short and should be extended to outline the content of the review. What was the purpose to write this review and who is the target audience? The authors could summarise in the introduction the key hypotheses and overall findings.
We now extended the introduction following this guidance.
2) In line 37 references should be given for debate about human adult neurogenesis. For example Kempermann et al., 2018 Cell Stem Cells.
Thanks for this good suggestion. We have now added that reference, as well as the original two articles that review focused the controversy on (Boldrini et al 2018; Sorrells et al 2018). It was very interesting to read these, as similar discrepancies were found in Drosophila, as we discussed in our article. Thank you.
3) In parts the manuscript could benefit from a revised structure. Especially in section 2 it is not clear why certain neural stem/progenitor types are introduced and others are not. It makes sense to describe the different modes of divisions and terminations of Type IA /ID ,Type II and Mushroom body neuroblasts. However, it is not clear why Type III neuroblasts are mentioned and other stem cells types like OPC and IPC neuroepithelial cells, lamina precursors and others are not described. I would focus and describe in detail the types of precursors, the reader needs to know to understand the subsequent sections. Here, an additional illustration that shows the two main modes of termination (Apoptosis for Type IA, MB neuroblasts and terminal division for Type ID neuroblasts) would be informative. See for example Maurange and Gould, 2005, Trends in Neurosciences.
We are grateful to the Reviewer for pointing out that that section could do with improvement. We agree that it was important to cover the neuroblasts that we would discuss in subsequent sections. Thus, as we discuss multiple findings of neurogenesis in central brain and optic lobes, we felt it would not be right to exclude these neuroblasts. Rather, it would be best to expand on them. Thus, we have revised the neuroblast literature, and now we include multiple new references. We have improved the description of and details about the different types of neuroblasts in the brain. All section 2, from line 62 to 144 has been rewritten. Please see the manuscript with the tracked changes, which we are also submitting.
Following the advice of this Reviewer, we have also made a new figure (new Figure 1), which illustrates the larval NBs, and their elimination either through cell cycle exit or apoptosis.
4) A crucial open question concerning adult neurogenesis in general could be discussed in a concluding section. Is adult neuro/gliogenesis in Drosophila functionally relevant for brain plasticity and repair? What do we know from other (invertebrate) model organisms ?
We have now expanded on the concluding paragraph. In the absence of further evidence, we prefer not to speculate further.
5) The manuscript could benefit from editing by a proficient English speaker. Structure and logic in language are not always coherent.
Our manuscript has been read by two British colleagues (Elizabeth Connolly and Jacob Hasenauer). We have made changes throughout, please see the submitted tracked manuscript highlighting all the changes. We hope the improvements are to the Reviewer’s satisfaction.
Minor points to address:
1) Line 28-35: should be removed.
done
2) Gene names should be written in full at the first time and subsequently abbreviated. For example see line 362.
done
3) In line 182 it should read:…..after incorporating EdU.
corrected
4) Line 191: Detecting mitosis in the adult Drosophila brain is difficult.
changed
5) Line 392: Delete: 5.
done
Reviewer 2 Report
This is a comprehensive and balanced review about neurogenesis in the adult Drosophila brain. I found this a very interesting read, and the authors provide a nice case to support the existence of neurogenesis in the adult Drosophila brain - covering both the evidence for and against in a balanced way that discusses the caveats of various experimental approaches used to test this. The figures and tables are well presented and will be useful to researchers in the field. Minor point - some of the instructions to authors have been left in the text (lines 28 - 35 introduction, 414 - 409 conflicts/acknowledgements) and should be removed from the final document.
Author Response
Reviewer 2
This is a comprehensive and balanced review about neurogenesis in the adult Drosophila brain. I found this a very interesting read, and the authors provide a nice case to support the existence of neurogenesis in the adult Drosophila brain - covering both the evidence for and against in a balanced way that discusses the caveats of various experimental approaches used to test this. The figures and tables are well presented and will be useful to researchers in the field. Minor point - some of the instructions to authors have been left in the text (lines 28 - 35 introduction, 414 - 409 conflicts/acknowledgements) and should be removed from the final document.
We are very pleased that this Reviewer only had positive comments about our manuscript. We have now removed those ‘instructions to authors” sentences.
Round 2
Reviewer 1 Report
The authors have addressed all my comments and improved the structure and readability of the manuscript. Several paragraphs of the manuscript were rewritten. An additional Figure illustrating the the different neural precursor types in the larval brain was added. I can recommend publication in its present form.
Minor comments:
line 165: anti-Deadpan (Dpn)
line 181: brain tumour (brat)